# Telocytes regulate macrophages in periodontal disease

**Jing Zhao[1,2], Anahid A Birjandi[1], Mohi Ahmed[1], Yushi Redhead[1], Jose Villagomez Olea[1], Paul Sharpe[1]***

[1]Centre for Craniofacial and Regenerative Biology, Faculty of Dentistry, Oral and Craniofacial Sciences, King's College London, London, United Kingdom; [2]Department of Oral and Maxillofacial Implantology, Shanghai Ninth People's Hospital, Shanghai Jiao Tong University School of Medicine, Shanghai, China

**Abstract** Telocytes (TCs) or interstitial cells are characterised in vivo by their long projections that contact other cell types. Although telocytes can be found in many different tissues including the heart, lung, and intestine, their tissue-specific roles are poorly understood. Here we identify a specific cell signalling role for telocytes in the periodontium whereby telocytes regulate macrophage activity. We performed scRNA-seq and lineage tracing to identify telocytes and macrophages in mouse periodontium in homeostasis and periodontitis and carried out hepatocyte growth factor (HGF) signalling inhibition experiments using tivantinib. We show that telocytes are quiescent in homeostasis; however, they proliferate and serve as a major source of HGF in periodontitis. Macrophages receive telocyte-derived HGF signals and shift from an M1 to an M1/M2 state. Our results reveal the source of HGF signals in periodontal tissue and provide new insights into the function of telocytes in regulating macrophage behaviour in periodontitis through HGF/Met cell signalling, which may provide a novel approach in periodontitis treatment.

*For correspondence:
paul.sharpe@kcl.ac.uk

Competing interest: The authors declare that no competing interests exist.

## Editor's evaluation

This article presents valuable findings on the role of a relatively understudied cell type, telocytes, in a mouse model of periodontitis. Using single-cell RNA-seq and cellular assays, the authors present convincing evidence that telocytes signal to macrophages using HGF to shift their polarisation state from inflammatory (M1) to a more 'tissue-remodelling' state (M1/M2). Since periodontitis is linked to many other illnesses (e.g. rheumatoid arthritis, cardiac disease, Alzheimer's disease), new insights into the cell types that play a role in the progression of the disease are important to the field of inflammatory and chronic diseases. Future studies will need to elucidate whether telocytes play similar roles in their other niches.

## Introduction

Periodontitis is an inflammatory disease of the periodontal ligament (PDL), the tissue that connects teeth to alveolar bone. It is a prevalent, incurable, and continuous degenerative disease that results in bone loss and tooth loss. Individuals with periodontitis often exhibit gingiva recession, bleeding, and tooth mobility (*Dentino et al., 2013*). All periodontitis (with attachment loss) develops from gingivitis (no attachment loss) with poor prognosis but not all gingivitis develops into periodontitis.

Numerous studies have focused on the causes of periodontitis. Briefly, pathogens accumulate on the tooth surface (forming plaque), invade the periodontium tissue, and release lipopolysaccharide (LPS), which results in inflammation and immunological events. LPS causes the polarisation of pro-inflammatory macrophages (M1). M1 macrophages release cytokines such as TNF-α, IFN-γ, IL-6,

and IL-12, which contribute to the development and progression of inflammation-induced tissue destruction (*Zhou et al., 2019a*; *Huang et al., 2018*; *Viniegra et al., 2018*; *de Vries et al., 2017*). Hence, understanding the regulation of the inflammatory response is critical to understanding and treating periodontitis. The PDL, albeit only present as a thin layer, contains several different cell types. However, the understanding of cell populations in the PDL, their interactions, signalling pathways, and how these are impacted by disease and inflammation are poorly understood.

In this study, we describe a cell type not previously identified in the PDL, namely, telocytes (interstitial cells). Telocytes are an enigmatic interstitial cell type that are best characterised by their unusual morphology having very long projections that make direct contacts with other cells. They are believed to play a role in direct cell–cell communication establishing three-dimensional networks guiding tissue organisation, mechanical sensing, regulating immune responses, and phagocytic-like properties (*Cretoiu et al., 2017*; *Kondo and Kaestner, 2019*). Transmission electron microscopy (TEM) of telocytes shows extracellular vesicles bulging out from their membranes, suggesting active

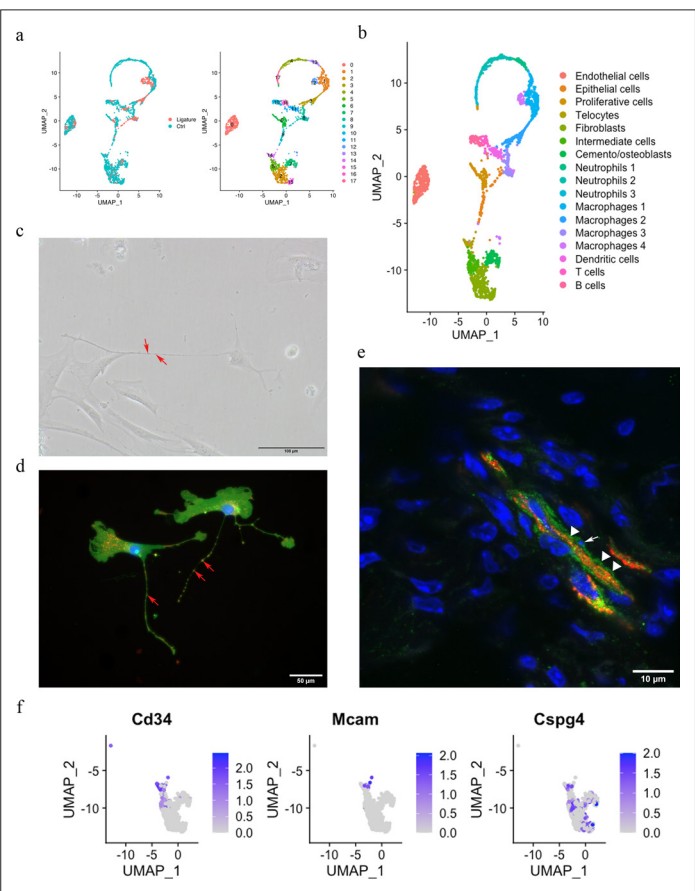

**Figure 1.** Telocytes in the periodontal ligament (PDL). (**a**) PDL single cells from control mice and ligature-treated mice were combined and clustered into 18 clusters. (**b**) Identification of each cluster. Telocyte clusters were identified by CD34+ and CD31-. Macrophages are in four clusters. (**c**) In vitro cell culture with CD1 PDL cells at passage 1 shows characteristic telocyte structure, including podoms (red arrows, the dilated portion) and podomers (between two arrows, the thin segments between podoms). (**d**) Wnt1 lineage-traced cells (GFP in green) were cultured and stained with CD34 in red, which show piriform cell body and moniliform podoms (red arrows) and podomers. (**e**) Telocytes (CD34+CD31-) was detected near blood vessel (CD34+CD31+) in vivo, CD34 in green, CD31 in red. White arrow indicates the small nuclei, and white triangle shows the elongation of telocyte respectively. (**f**) CD34 expression was compared with pericyte markers, CD146 (Mcam) and NG2 (Cspg4) expression.

The online version of this article includes the following figure supplement(s) for figure 1:

**Figure supplement 1.** Heatmap of cell clusters show telocytes distinct from other mesenchymal cells.

**Figure supplement 2.** Telocyte identification makers. Panel (**b**) shows a zoomed region of boxed region in (**a**).

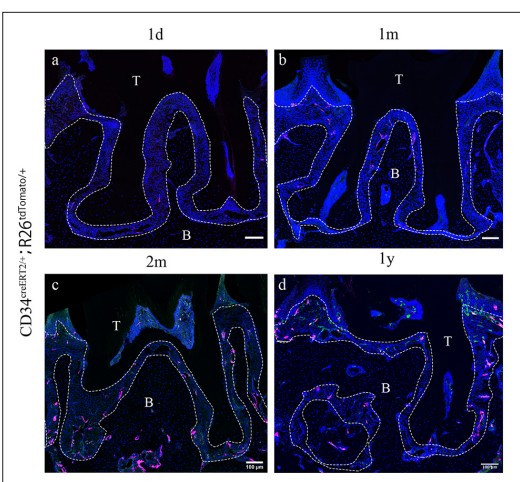

**Figure 2.** *Cd34^creERT2* lineage tracing showing limited contribution of telocytes to periodontal ligament (PDL) homeostasis in adulthood. In the adult stage, *Cd34^creERT2/+; Rosa26^tdTomato/+* mice were used to trace from 7 weeks for 1 day (**a**), 1 month (**b**), 2 months (**c**), and 1 year (**d**). CD34 lineage-traced cells in red, CD31 were co-stained in green. CD34+CD31- cells are telocytes which were rarely found in periodontal tissue and randomly dispersed during homeostasis. Increase of CD34+CD31- cell number was not detected as the extending of tracing time. Scale bars = 100 µm. B, bone; T, tooth. Periodontal tissue in dotted line.

physical communication with other cells (*Cretoiu et al., 2016*). To date, possible cell signalling roles played by telocytes have not been fully described. It is only known that in the intestine subepithelial telocytes are identified as an important source of Wnt signals to maintain the proliferation of intestinal stem cells (*Shoshkes-Carmel et al., 2018*).

Telocytes are identified in tissues by dual immunolabeling, most commonly CD34+/CD31-, CD34+/c-Kit+, CD34+/Vim+, and CD34+/PDGFRα+ (*Kondo and Kaestner, 2019*; *San Martin et al., 2014*; *Li et al., 2016*; *Wang et al., 2020*; *Manetti et al., 2015*). In this study, we use CD34+/CD31- to identify PDL-resident telocytes combined with genetic lineage tracing and Sc-RNA sequencing to determine the role of telocytes in homeostasis and periodontitis. We show that quiescent telocytes located near blood vessels are activated in periodontitis and regulate macrophages via the hepatocyte growth factor (HGF)/Met signalling pathway. The resulting transition of macrophages from an M1 to M2 state provides a possible therapeutic strategy for treating periodontitis.

## Results

### ScRNA-seq analysis reveals a telocyte population in PDL

The PDL is made up of both neural crest-derived and mesodermal-derived cell types, and we have previously described the constituent cell populations using single-cell transcriptomics (*Wang et al., 2020*). This analysis compared adult PDL in homeostasis with PDL tissue from a ligature-induced periodontitis mouse model (*Zhao et al., 2021*). The two datasets were integrated by performing a canonical correlation analysis (CCA) (*Zhao et al., 2014*) identifying 2270 cells for analysis after filtering. These cells were further divided into 18 unsupervised clusters for annotation (*Figure 1a*). Differential expression (*Figure 1—figure supplement 1*) revealed cell clusters, including endothelial cells (*Pecam1*), epithelial cells (*Krt14, Krt5*), B cells (*Cd79a, Cd79b*), T cells (*Skap1, Trac*), dendritic cells (*Cd209a, Mg12*), macrophages (macrophage 1 [*Plek, Cd80*], macrophage 2 [*Wfdc17, Mpeg1*], macrophage 3 [*Apoe, Cxcr1*], macrophage 4 [*Arg1, Cd36*]), mast cells (*Cmal, Tpsb2*), neutrophils (neutrophil 1 [*Retnlg, Mmp8, Mmp9*], neutrophil 2 [*Ngp, Cd177, Chil3*], neutrophil 3 [*Fcnb, Chil3*]), a proliferative cell population (*Stmn1, Mki67, Pclaf*), cemento/osteolineage cells (*Ibsp, Bglap, Spp1*), fibroblasts (*Postn, Aspn*), and an intermediate cluster between clusters 2, 15, and 14. A small population of cells (cluster 14) were found in the mesenchymal cell population that expresses *Cd34*; however, they did not express the endothelial cell marker *Cd31*, and we thus identified these cells as telocytes (TCs) (*Figure 1b*).

According to the literature, telocytes can be identified by dual immunolabeling, most commonly CD34+/CD31-, CD34+/c-Kit+, CD34+/Vim+, and CD34+/PDGFRα+ (*Kondo and Kaestner, 2019*; *San Martin et al., 2014*; *Li et al., 2016*; *Wang et al., 2020*; *Manetti et al., 2015*). However, in our ScRNA sequencing data, we found that the expressions of *Kit, Vim,* and *Pdgfrα* are either low with not much overlap with *Cd34* or ubiquitous (*Figure 1—figure supplement 2*). To confirm these cells as telocytes, we first cultured PDL cells in vitro and searched for cells with a typical telocyte morphology. Typical telocyte cell morphology was observed with small cell bodies and a long cellular process called telopodes (*Figure 1c*). Telopodes consist of dilated portions (podoms) and thin segments in between (podomers). To determine whether the telocyte-like cells are derived from neural crest, we collected PDL cells from *Wnt1^Cre/+;Rosa26^mTmG/+* mice, which labels neural crest-derived cells (*Graves et al.,*

*2008*) and stained for GFP and CD34. GFP-positive cells showed telocyte structures with podoms and podomers, which were also positive for CD34 (*Figure 1d*). To identify telocyte locations in situ, we co-immunostained sections with CD34 and CD31. As shown in *Figure 1e*, telocytes (CD34+/CD31-) were found in close association with blood vessels (CD34+/CD31+), had small cell nuclei, and long cell protrusions. However, pericytes are also in close proximity to blood vessels. Therefore, to discriminate between telocytes and pericytes, we compared genes that are expressed in pericytes, *Cd146* and *Ng2*. We found that these genes were not expressed in telocytes (*Figure 1f*). These data collectively suggest that CD34+/CD31- telocytes are present in the PDL located in the vicinity of blood vessels.

## Quiescent telocytes are activated in periodontitis

Since telocytes only account for a small subset of cells in the RNAseq datasets in PDL during homeostasis together with the big overall increase in immune cells in the disease datasets, thereby masking any changes in telocyte numbers, we addressed whether these cells are quiescent or actively proliferating during homeostasis directly on tissue sections. Lineage tracing using *Cd34^{creERT2/+};Rosa26^{td-Tomato/+}* mice (*Jiang et al., 2021*) followed by a 1 day–1 year post-tamoxifen chase period revealed that CD34+/CD31- cell numbers did not increase to any significant extent (*Figure 2*), suggesting that telocytes are a small, quiescent cell population in homeostasis. To investigate whether telocytes responded to disease, we used our established ligature-induced periodontitis mouse model where sutures are placed around the second molars (*Wang et al., 2020*). The ligature leads to plaque accumulation and thus facilitates the invasion of bacteria (*Nakamura et al., 1984*). By measuring the distance between alveolar bone crest and cemento-enamel junction (ABC-CEJ distance) (*Figure 3a*), we found that bone loss reached a maximum between days 4 and 7 for all three molars (*Figure 3b*). Notably, even though only the second molar was subjected to a ligature, the first and third molars also showed some bone loss at early time points, suggesting that they are affected by the ligature-induced periodontitis to some extent. However, the first and third molar bone loss was recovered at longer time points (*Figure 3b*), indicating the self-recovery ability from milder periodontitis.

We visualised telocytes with CD34 and CD31 antibodies in ligature PDL at different time points. After 2 hr, there was an obvious increase in the number of CD34+/CD31- telocytes close to blood vessels (*Figure 3c*, *Figure 3—figure supplement 1*). *Cd34^{creERT2/+};Rosa26^{tdTomato/+}* mice (*Jiang et al., 2021*) tracing for 7 days also showed an increase in tdTomato+/CD31- cells in periodontitis (*Figure 3d and e*, *Figure 3—figure supplement 2*). These telocytes also expressed the proliferation marker Ki67 (*Figure 3f*). These data indicate that telocytes proliferate in the PDL following ligature-induced periodontitis.

It has been demonstrated that telocytes can secrete extracellular vesicles (*Cretoiu et al., 2016*), suggesting that they may have a role in cell signalling. Gene enrichment analysis of the single-cell RNA-seq datasets identified angiogenesis, leukocyte migration and inflammatory responses as the three top pathways in PDL telocytes (*Figure 3g*).

## Telocytes regulate macrophages via HGF/Met signalling pathway

To functionally understand the differences between homeostasis and periodontitis, we interrogated the RNA-seq datasets to compare the two conditions with respect to cell–cell communication pathways. We identified the CHEMERIN, HGF, IFN-I, IL16, LIFR, and APELIN pathways as not being active during homeostasis but rather, to be active in periodontitis (*Figure 4a*). Telocytes were found to express *Hgf* and *Flt3* (*Figure 4b*), highlighting a potential role for the HGF signalling pathway.

HGF was originally found in liver as a potential hepatocyte mitogen (*Nakamura et al., 2000*). It is involved in repair and regeneration as a healing factor (*Matsumoto et al., 2014*; *Ko et al., 2018*; *Yang and Ming, 2014*). In mouse ligature-induced periodontitis, our RNA-seq cell–cell communication analysis identified macrophages as a target of HGF signalling from telocytes (*Figure 4c*). Of the four macrophage clusters identified, telocytes are identified as potentially interacting with macrophage clusters 1 and 4. To confirm that telocytes express HGF, triple immunofluorescence staining was performed with CD34, CD31, and HGF. Endothelial cells (CD34+/CD31+) did not express HGF, whereas telocytes (CD34+/CD31-) with telopodes expressed HGF (*Figure 4d*). This is consistent with Sc-RNA seq analysis (*Figure 4e and f*). In conclusion, telocytes express HGF and, based on the ScRNA-seq analysis, are the only cells that produce this signal in periodontitis. The recipient cells are macrophages which express the HGF receptor (HGFR), encoded by *Met* (*Figure 4c*).

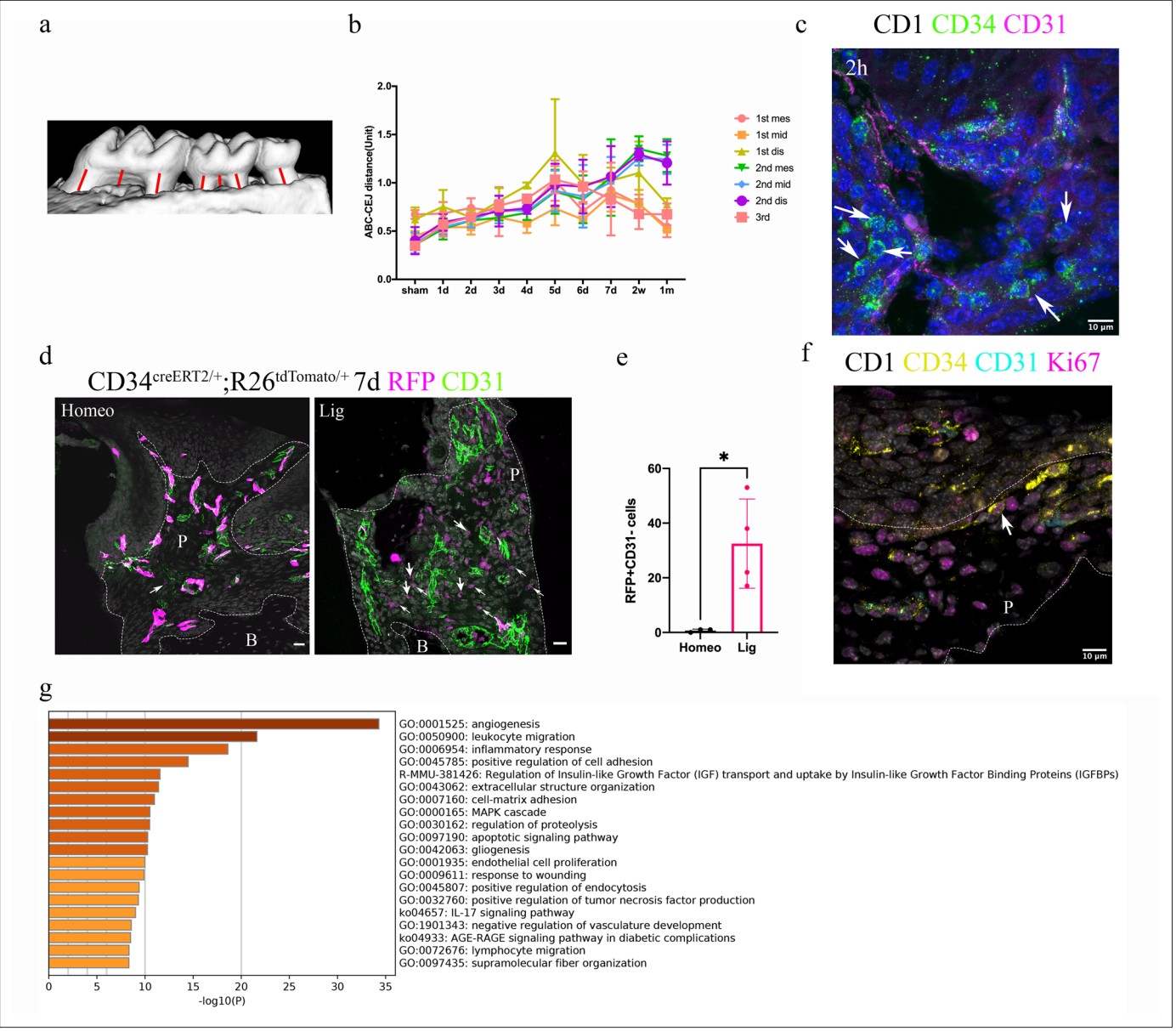

**Figure 3.** Telocytes are activated in response to periodontitis by increasing their number. (**a**) Illustration of bone loss measurement. For the first and second molars, the ABC-CEJ distance of mesial and distal roots in parallel to the root long axis and the ABC-CEJ distance in the trifurcation area were measured; for the third molar, the ABC-CEJ distance in the middle of the tooth was measured. (**b**) Quantification from micro CT results indicates the change of bone loss in periodontitis plotted by time course (n = 3). Hard tissue around the second molar is severely affected, the time for all molars reaching the bone loss plateau is between day 5 and day 7. 1 unit = 0.4 mm (n = 3). (**c**) An accumulation of telocytes was found in periodontitis as early as 2 hr after the ligation procedure. These telocytes (CD34, green) were mostly found around blood vessels (CD31, red), especially tissue toward the crown. Scale bars = 10 μm. (**d**) *Cd34^{creERT2/+}; Rosa26^{tdTomato/+}* mice were given three tamoxifen injections started from the procedure day and harvested on day 7. Significantly increased CD34+ cells (red) were observed in the periodontitis group. There were more endothelial cells (green) observed on day 7 of periodontitis. The telocyte (red)-derived cells were not overlapping with endothelial cells (green). Scale bars = 20 μm. (**e**) Statistical analysis of numbers of telocytes comparing homeostasis (n = 3) and periodontitis (n = 4). (**f**) Telocytes (CD34+, CD31-) express a proliferation marker, Ki67 in periodontitis. CD34: yellow; CD31: cyan; Ki67: magenta; nuclei: grey. (**g**) 863 input genes highly expressed in telocytes cluster with avg_logFC > 0 were selected for gene enrichment analysis. The 20 best p-value terms are plotted. The bar plot is coloured by p-values. B, bone; P, periodontal tissue.

The online version of this article includes the following figure supplement(s) for figure 3:

**Figure supplement 1.** An increased number of telocytes was observed on days 2 and 4 post procedure.

**Figure supplement 2.** Telocytes show lower *Cd34* expression compared to endothelial cells.

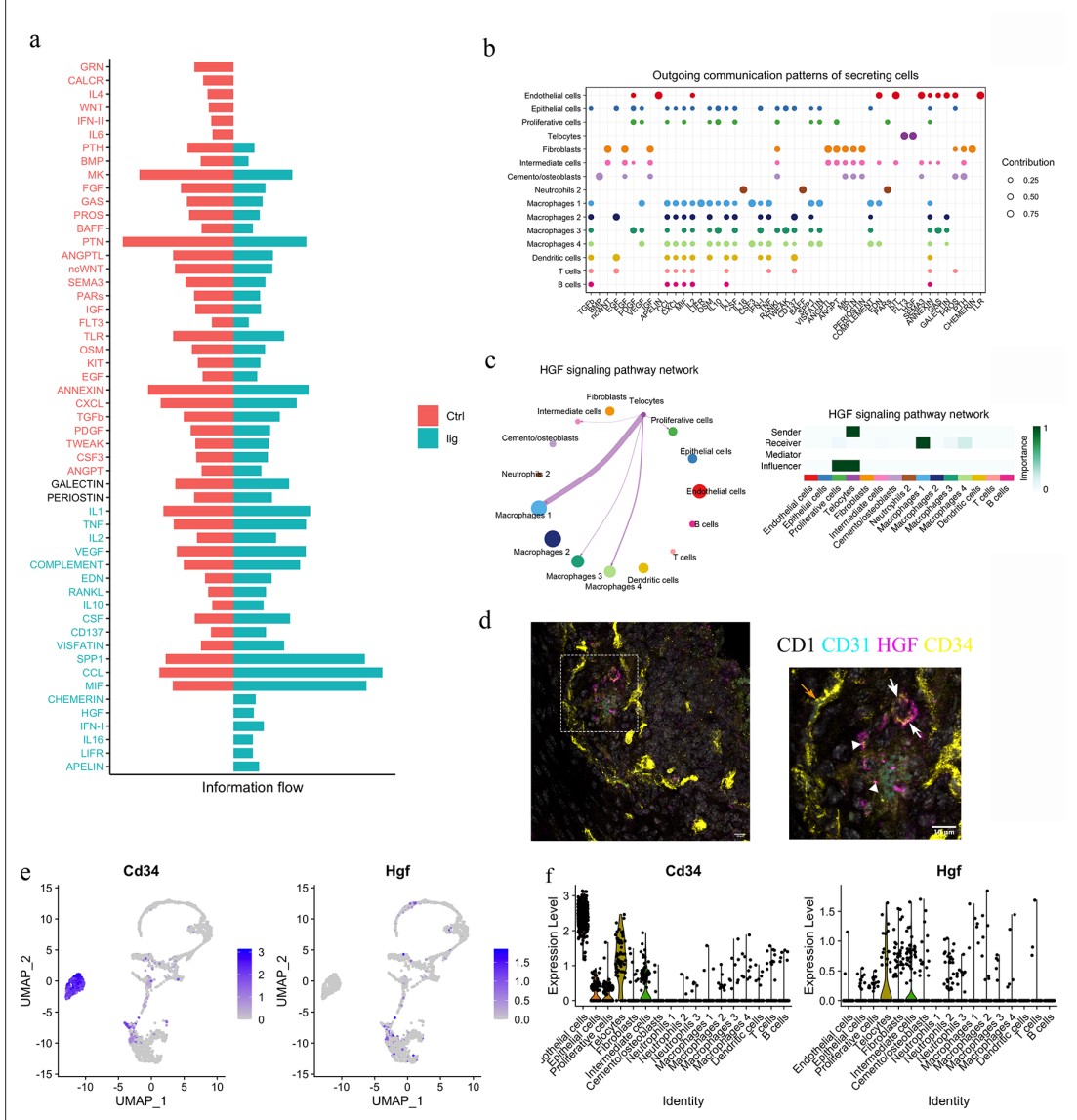

**Figure 4.** Telocytes regulate macrophages via HGF/Met signalling pathway. (**a**) Comparison analysis shows HGF pathway is upregulated in periodontitis (**b**) Cell–cell communication analysis was performed on disease dataset based on secreted signals database. The outcoming patterns were plotted. Bubble plot suggests that telocytes send signal molecules in FLT3 and HGF signalling pathways exclusively in periodontitis (dark purple dots). (**c**) Circle plot and heatmap suggest telocytes send HGF signals to macrophage clusters 1, 3, and 4 in periodontitis. (**d**) Immunostaining on CD1 mice periodontitis tissue for CD34+/CD31- cells indicates telocytes (white arrows) and CD34+/CD31+ cells for endothelial cells (orange arrow) from CD1 mice. The typical morphology of telocytes, podoms are denoted with arrowheads. The telocytes were expressing HGF (magenta). Scale bars = 10 μm. (**e**) Expression of CD34 (marker of telocyte) and Hgf were both found in the telocyte cluster and some cells in the intermediate cell cluster. (**f**) CD34 is expressed in endothelial cells and telocytes, and Hgf is expressed in telocytes but not endothelial cells and some intermediate cells that are close to telocytes.

## Macrophages receiving HGF signals show M1–M2 transition

Four clusters of macrophages were identified from the single-cell transcriptomics analysis (**Figure 5**). Differentially expressed gene (DEG) analysis suggested macrophage cluster 1 express *Acod1*, *Trem1*; macrophage cluster 2 express *Mc4a4c*, *Ccr2*; macrophage cluster 3 express *Aif1*, *Mrc1*; macrophage cluster 4 express *Cd36*, *Arg1* (**Figure 5a**). To gain a better understanding of these cells, we performed unsupervised clustering on the four clusters. In each re-clustered plot, clusters on the left are cells from periodontitis and right from homeostasis – they are clearly separated and the periodontitis cells occur in three linked clusters (0, 3, and 6) (**Figure 5b, e, and f**).

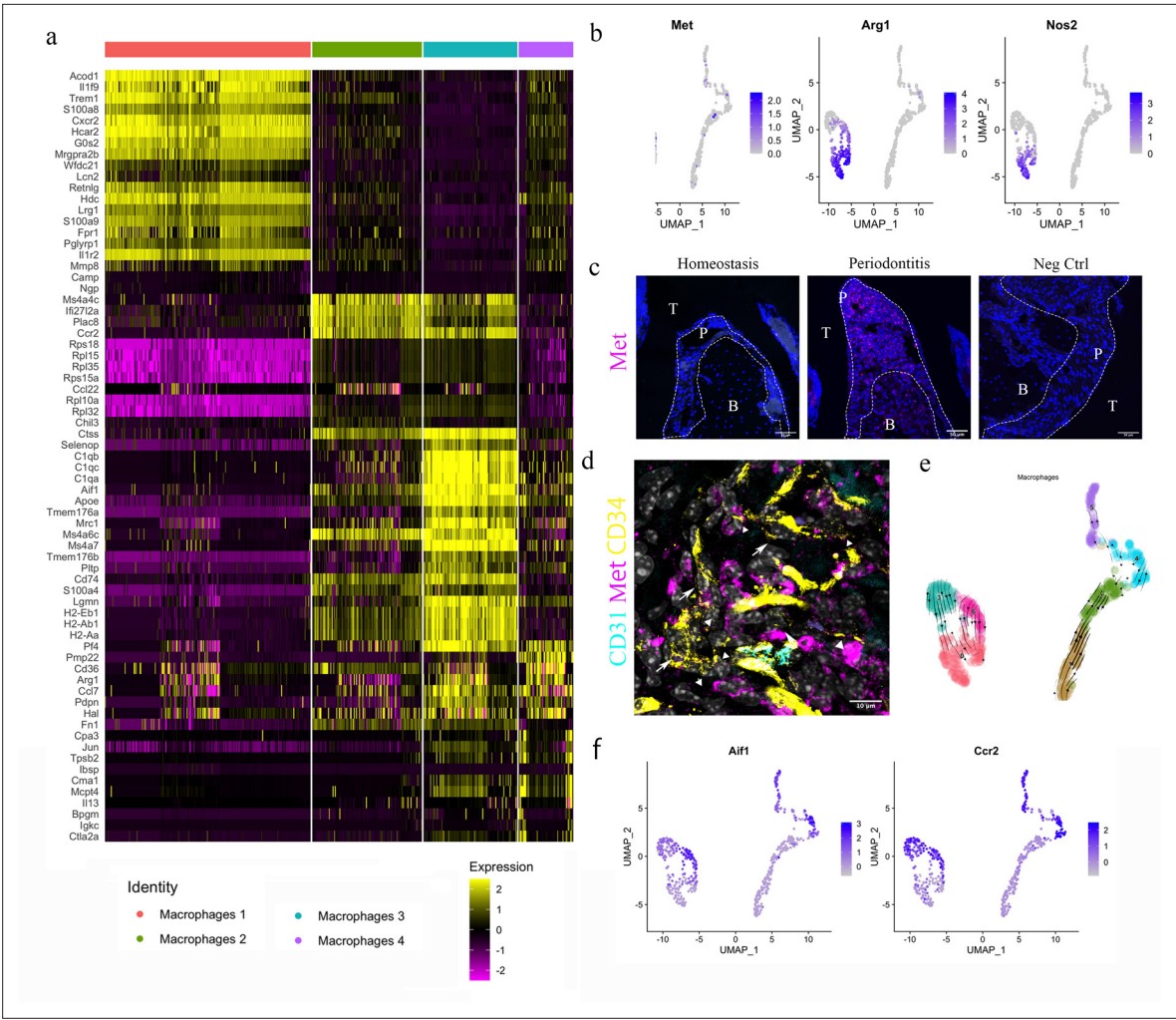

**Figure 5.** HGF/Met signalling drives M1 to M2 transition. (**a**) Heatmap presenting the differentially expressed gene (DEG) of four macrophage subpopulations (top 20 DEG genes). (**b**) Macrophages were extracted from the complete dataset and re-clustered. In a feature plot, all the macrophages in disease dataset are in the left cell cluster, separated from macrophages in homeostasis (cells on the right). Feature plots show Met-expressing macrophages express M1 marker *Nos2* and M2 marker *Arg1*. (**c**) Met protein was not detected in periodontal ligament (PDL) homeostasis but in periodontitis. Met: magenta; nuclei: blue. Scale bars = 50 μm. (**d**) Telocytes (yellow, indicated by arrows) are making contact with Met-expressing cells (magenta) by using their protrusions (telopodes indicated by arrowheads). CD34: yellow; CD31: cyan; Met: magenta; nuclei: grey. Scale bar = 10 μm. (**e**) RNA velocity shows *Met+* macrophages (cluster 0) are related to those in clusters 3 and 6. (**f**) Macrophages expressing *Met* (cluster 0) are related to those in cluster 6 (*Ccr2+Aif1*^hi^) and cluster 3 (*Ccr2+ Aif1*^lo^) macrophages.

The online version of this article includes the following figure supplement(s) for figure 5:

**Figure supplement 1.** Met expression in homeostasis and periodontitis.

The HGF signalling pathway has a sole ligand–receptor pair, HGF and HGFR (Met). Interestingly, in the datasets, macrophages expressing Met also expressed *Arg1* (encodes Arginase1 [Arg1]) and *Nos2* (encodes inducible nitric oxide synthase [iNOS]) (*Figure 5b*).

In periodontitis, iNOS mediates the pathological effect of LPS and it is a marker of inflammatory M1 macrophages, which are an indicator of disease progression (*Italiani and Boraschi, 2014*). *Arg1*, a gene expressed in M2 macrophages, however, is believed to decrease LPS-induced pro-inflammatory cytokine production (*Italiani and Boraschi, 2014*). Therefore, macrophages activated by HGF signals in the PDL can be considered to be in an M1/M2 state. Previous studies indicate that monocytes in an inflammatory environment first polarise to M1, then M2 subject to microenvironmental signals (*Yang et al., 2018*; *Bronte and Zanovello, 2005*). It is well known that Arg1 competes with iNOS for their common substrate L-arginine (*Italiani and Boraschi, 2014*; *Nishikoba et al., 2020*). Moreover, HGF/

Met is reported to induce *Met+/Nos2+* macrophages to an M2-like phenotype via overexpression of *Arg1* (*Nishikoba et al., 2020*).

During homeostasis, macrophages in PDL do not express Met protein; however, macrophages start to express Met in periodontitis (*Figure 5c*, *Figure 5—figure supplement 1*). Met-expressing cells are located in close proximity to the telopodes of telocytes (*Figure 5d*), indicating that telocytes likely make physical contact with Met+ macrophages. Thus, it is conceivable that telocytes promote the overexpression of *Arg1* in periodontitis which further leads to the M1/M2 state of macrophages. Additionally, RNA velocity analysis suggests that these *Arg1+/Nos2+* macrophages present in cluster

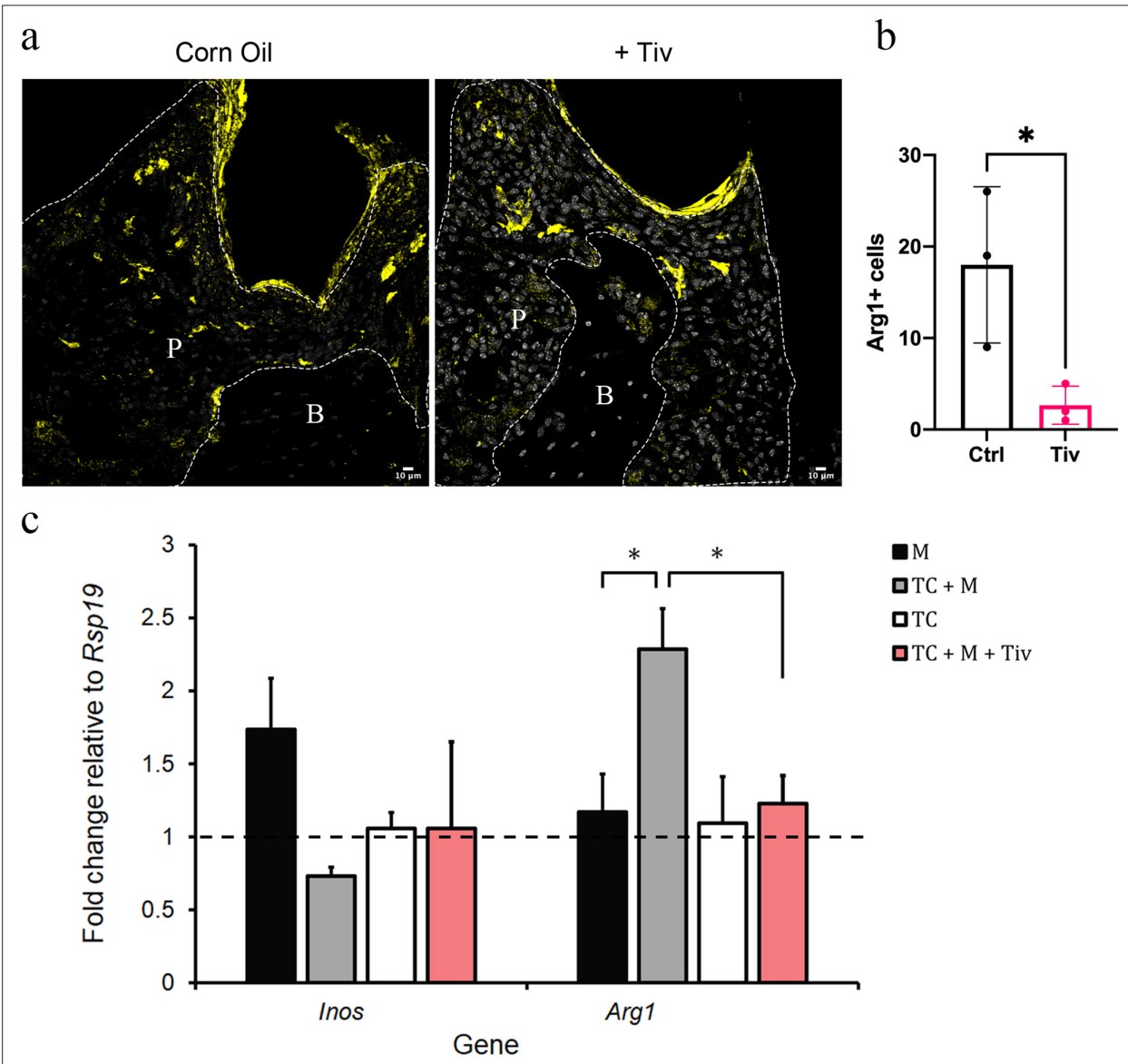

**Figure 6.** Telocytes promote M1 to M2 transition via HGF/Met signalling. (**a**) CD1 mice were used to induce periodontitis. Corn oil or tivantinib (Tiv) were given once on day 5 post procedure. Samples were collected 12 hr after drug delivery. Mice given corn oil (left) show more cells expressing Arg1+ cells (yellow) in periodontium than the mice given tivantinib (right). Scale bars = 10 μm. (**b**) Statistical analysis shows significant difference in Arg1 expression between control group (corn oil) and tivantinib-treated group (p<0.05). (**c**) qPCR data showing FAC-sorted telocytes (TC) cultured with M0/1 macrophages (M) in the presence of LPS leads to an increase in the M2 marker *Arg1*, with simultaneous reduction in the M1 marker *Inos*. Transition of M1 to M2 is significantly reduced upon addition of the HGF/Met inhibitor tivantinib. Error bars represent the standard error of mean (p<0.05). B, bone; T, tooth; P, periodontal tissue.

The online version of this article includes the following figure supplement(s) for figure 6:

**Figure supplement 1.** Macrophage polarisation marker gene expression in cultured M1 macrophages and FAC-sorted telocytes.

0 (*Figure 5b*) are linked with *Ccr2*hi*Aif1*lo (cluster 3) cells and *Ccr2*lo*Aif1*hi cells (cluster 6) (*Figure 5e and f*).

To experimentally determine whether *Arg1* expression in macrophages is increased by HGF signals from telocytes, tivantinib (ARQ197), a small-molecule Met inhibitor, was used to inhibit HGF/Met signalling. Tivantinib is a highly selective, non-ATP-competitive, orally available inhibitor of Met (*Aoyama et al., 2014*). We observed that when tivantinib was administered to mice with ligature-induced periodontitis, Arg1+ cell numbers significantly decreased (*Figure 6a and b*), indicating that inhibition of the HGF-Met signalling interaction in macrophages blocks their polarisation.

To demonstrate that telocytes are a source of HGF signals for macrophage transition from M1 to M2, we FAC-sorted CD34+/CD31- telocytes and co-cultured these with M0/M1 macrophages. It is important to point out that the CD34+/CD31- expression profile is currently the best identity marker for telocytes described in the literature (*Kondo and Kaestner, 2019*; *San Martin et al., 2014*; *Li et al., 2016*; *Wang et al., 2020*; *Manetti et al., 2015*) but other undetermined cells also have this expression profile. The FAC-sorted cells are a heterogenous population as not all adopted a characteristic telocyte morphology following 24 hr in culture. Nonetheless, we obtained sufficient cells with telocyte morphology. For better characterisation of macrophage polarisation markers, we noted that the literature describes a number of markers including *Nos2* (*Inos*) and *Cd80* as M1 markers, and, *Arg1* and *Cd163* as M2 markers (*Jablonski et al., 2015*). However, only *Inos* and *Arg1* are consistently reliable in our qPCR experiments as highly expressed in macrophages compared to telocytes (*Figure 6—figure supplement 1*). Therefore, we used *Inos/Arg1* as M1/M2 polarisation markers for subsequent qPCR analyses. Upon analysing the expression levels of *Inos* and Arg1 by qPCR, in comparison to control macrophage-only cultures, the presence of telocytes led to a significant increase in *Arg1* expression with simultaneous reduction in *Inos* expression (*Figure 6c*). However, this did not occur when tivantinib was added to the culture (*Figure 6c*). Together, these data suggest that M1 macrophages respond to HGF signals secreted by telocytes via Met to adopt an M1/M2 phenotype.

## Discussion

Based on in vitro, in vivo, and in silico studies, we identified a neural crest-derived cell population by morphology and expression of CD34+/CD31- in mouse periodontium. These cells, telocytes, are in a quiescent state in normal periodontal tissue unless challenged by periodontitis whereupon they increase in number and secrete HGF. Although we did not detect an obvious increase in telocyte numbers in our periodontitis sequencing dataset, this was likely a result of the overall relative decrease in stromal cell numbers caused by the increase in immune cells. In periodontitis, our sequencing data showed a larger proportion of inflammatory cells and relatively fewer stromal cells, indicating the successful induction of the inflammatory disease. In periodontitis, the emergence of a new macrophage population not present in homeostasis, together with signalling changes in telocytes, raised the possibility of a possible interaction between telocytes and macrophages.

We observed telocytes, as a major source of HGF, making physical contact with macrophages. Cleaved HGF can activate Met and downstream signalling pathways (*Joosten et al., 2020*). It is believed that pro-HGF, the inactive precursor, is secreted and processed by HGF activator (HGFAC), a zymogen in the circulation to achieve its function (*Fukushima et al., 2018*). Macrophages receiving HGF signals express *Met*, *iNos,* and *Arg1*, representing an M1/M2 state. The *Arg1+*/*Nos2+* macrophages have been recently proposed to exist in other tissues (*Locatelli et al., 2018*; *Mould et al., 2019*; *Bronte et al., 2003*). These cells do not exist in homeostasis but are detected in periodontitis identifying HGF/Met as a key macrophage regulatory pathway. Our findings are consistent with reports that LPS causes M1 polarisation and a shift towards M2 polarisation mediated by HGF signals (*Choi et al., 2019*). Thus, expression of two competing enzymes, iNOS and Arg1, in Met-expressing cells results from regulation by telocytes, which shifts M1 macrophages to an M2 phenotype, resulting in an M1/M2 state. The transition can be effectively inhibited by an HGF/Met inhibitor, tivantinib. It is reported that the transition from LPS-induced M1 macrophages to M2 macrophages is controlled by PI3K or CaMKKβ-AMPK signalling pathway in Met-expressing cells via induction of Arg1 expression (*Nishikoba et al., 2020*; *Choi et al., 2019*). We propose the underlying mechanism is possibly the activation of HGF/Met signalling pathway that triggered the activation of PI3K or CaMKKβ-AMPK signalling pathway in macrophages.

The presence of M1 macrophages can cause bone loss, whereas M2 macrophages can help prevent bone loss in the PDL. Telocytes showed the ability to shift M1 to M2, indicating that HGF secreted by telocytes should be beneficial in reducing bone loss. By comparing the bone loss of the second molars with the first molars and third molars from 1 day to 1 month post ligature treatment, we found that only the molars without ligature treatment showed an ability to recover bone loss caused by periodontitis. Second molars, which have the ligature throughout, showed consistent bone loss, indicating any ability of telocytes to reduce bone loss is limited by constant physical stimuli, supporting the importance of maintaining oral hygiene in future clinical applications.

Ligature-induced periodontitis is considered as an appropriate model to mimic human periodontitis (*Abe and Hajishengallis, 2013*). However, whether telocytes are able to control the progression of human periodontitis requires further investigation. From the aspect of this study, an optimistic outcome is expected given the activation of telocytes or HGF/Met pathway under careful maintenance of oral hygiene. Coincidentally, exogenous application with HGF was found to improve periodontal bone regeneration in swine (*Cao et al., 2015*).

Additionally, telocytes may have a role in angiogenesis as shown in the gene enrichment terms (*Figure 3e*). More CD31+ cells were noticed in periodontitis samples (*Figure 3d*). Vascularisation is considered important for periodontal regeneration (*Liu et al., 2019*). Therefore, the role of telocytes in periodontitis may not only be the regulation of macrophages through HGF/Met signalling pathway but also through angiogenesis. Telocytes are the cells that provide niche signals in the intestine (*Shoshkes-Carmel et al., 2018*). Future work may focus on the signals that telocytes send to adjacent niche cells including stem cells (*Popescu et al., 2011*), endothelial cells (*Rosa et al., 2020*), and nerve cells (*Popescu et al., 2011*).

Collectively, our study demonstrates for the first time that telocytes increase in number in periodontitis and communicate with immune cells to positively regulate periodontitis via HGF. The activation of HGF/Met signalling pathway or the exogenous use of activated telocytes may be a promising therapeutic measure against periodontitis. This function of telocytes may also present in other inflammatory diseases where telocytes exist such as arthritis (*Rosa et al., 2018*). Furthermore, our study also has implications for cancer research, where telocytes were found present and HGF/Met signalling were found essential for cancer metastasis (*Arnold et al., 2017*, *Díaz-Flores et al., 2021*).

## Materials and methods

### Mice

All mice were maintained in the Biological Service Unit, New Hunts House, King's College London. Mice were exposed to a 12 hr light–dark cycle and with food and water available ad libitum. Wild-type CD1 mice were obtained from CRL (Charles River Laboratory, UK), $Wnt1^{Cre/+};Rosa26^{mTmG/+}$ mice (*Graves et al., 2008*) were from JAX 003829 and 007576, respectively. The $Cd34^{creERT2/+}; Rosa26^{tdTomato/+}$ mouse (*Jiang et al., 2021*) was a kind gift from Prof. Qingbo Xu (King's College London) (*Zhao et al., 2021*). Three intraperitoneal injections of tamoxifen were given at a dose of 2 mg/30 gbw (Sigma, T5648) for three consecutive days. Mice were sacrificed by exposure to a rising concentration of carbon dioxide or cervical dislocation followed by tissue dissection and tissue processing. All mouse work was approved by UK Home Office under the project license 70/7866 and P5F0A1579, approved by the KCL animal ethics committee.

### Animal disease model

Animals older than 8 weeks were used to induce periodontitis. Mice were anaesthetised with Ketavat and Domitor, injected 10 mL/kg i.p. The ligature procedure was performed as described (*Fleming et al., 2019*). Briefly, 5-0 wax-coated braided silk suture (COVIDIEN, S-182) was tied around the upper second molar in order to induce periodontitis. Samples were collected at desired time points.

### HGF/Met pathway inhibition

CD1 mice were used to induce periodontitis. Tivantinib (130 mg/kg in corn oil with 2.5% DMSO) was orally applied on day 5 post procedure. A control group was given corn oil with 2.5% DMSO. Samples were collected 12 hr later (n = 3).

## Immunofluorescence

Maxillae were dissected and fixed in 4% PFA overnight. Samples were decalcified in 19% EDTA until soft enough to cut (~7 days). Processed samples were then dehydrated with 30% sucrose followed by embedding in OCT on dry ice with ethanol. Cryosections were fixed by 4% PFA. Sections were then subject to permeabilisation by 0.2% Triton X-100 (Sigma, X100), heat-induced antigen retrieval, and blocking with 3% BSA. Sections were stained by the following antibodies: anti-RFP (Abcam, Ab62341), anti-CD34 (Abcam, Ab81289 and Ab8158), anti-CD31 (Abcam, Ab7388 and Ab24590), anti-GFP (Abcam, Ab13970), anti-Arg1 (Abcam, Ab92274), anti-Met (Abcam, Ab51067), anti-HGF (Abcam, Ab83760), and anti-Ki67 (Abcam, Ab16667). Secondary antibodies included Alexa Fluor 488 (Invitrogen, A11039), Alexa Fluor 568 (Invitrogen, A11077), Alexa Fluor 633 (Invitrogen, A21052), and Alexa Fluor 488 (Invitrogen, A11008). Tyramide signal amplification (NEL744001KT, PerkinElmer) was performed for weak signals. Hoechst 33342 (Invitrogen 62249, 1:500) was used for DNA staining. Slides were mounted using Citifluor AF1 (EMS, 171024-AF1) and cover-slipped for microscopy. Zeiss Apotome or Leica TCS SP5 systems was used for acquiring images. ImageJ and Adobe Photoshop were used for image processing.

## Single-cell RNA sequencing and analysis

For ScRNA-seq, adult CD1 mice were used. CD1 mice were sacrificed and dissected under a stereo-microscope with the gingiva carefully removed. Teeth were extracted and only the intact molars were kept. For periodontitis, only the second molars were used for subsequent use. The harvested molars were pooled and dissociated with 3 U/mL Collagenase P (COLLA-RO, Roche) followed by incubation for 45 min in a 37°C shaking water bath. The dissociation process was aided by dispersion with a 1 mL pipette every 15 min. Cells were then passed through a 40 µm strainer (Falcon 352340) followed by FACS for alive cells. Single cells in PBS with 0.04% ultrapure BSA were processed following a standard 10× genomic protocol (Chromium Single Cell 3' v3). Count matrices were generated from the fastq files via CellRanger pipeline using Ensembl 97 genome annotation. Ambient and background RNA from the count matrices were first removed using CellBender remove-background tool (*Satija et al., 2015*). Cells express >1000 features and with less than 20% mitochondria gene content were kept. A total of 2270 cells were used for analysis. Batch effect was removed by the Seurat (v3.2.0) CCA approach (*McInnes et al., 2018*). Integrated data were subsequently scaled and PCA was performed. Thirty dimensions were calculated based on variable features followed by UMAP (*Blondel et al., 2008*) for embedding and Louvain (*La Manno et al., 2018*) clustering (resolution 1) on knn graph. Macrophages (613 cells) were selected and re-clustered. RNA velocity data were generated using the velocyto tool (*Zhou et al., 2019b*). For gene enrichment analysis, metascape (https://metascape.org/gp/index.html#/main/step1; *Jin et al., 2021*) was used: genes highly expressed in telocytes cluster with avg_logFC > 0 were selected, and 863 input genes were used. Finally, cell–cell communication was estimated based on cell groups by using CellChat (v1.0.0) (*Stuart et al., 2019*), a method which provides a database that takes into account multi-subunit structures of ligand–receptor pairs, soluble agonists and antagonists, as well as membrane-bound co-receptors. The communication between cell types was analysed based on the secreted signalling database.

## In vitro studies

PDL cells from CD1 mice (n = 3) or *Wnt1^{Cre/+};Rosa26^{mTmG/+}* mice (n = 3) were collected for cell culture. Tissues were treated as above to harvest single-cell suspension. DMEM/F12 media (3:1) supplemented with 20% FBS, L-glutamine, and P/S were used for cell culture. Cells at passage 1 were used for analysis.

## Isolation of telocytes by FACS

Molar teeth and tongues were dissected in L-15 medium (Thermo Fisher, 21083027) from eight adult female mice for isolation and sorting of telocytes as CD34+/CD31- cells. Dissected pulp, PDL, and tongues were transferred into 1.5 mL tubes, excess L-15 was removed, and 100 µL of 20 U/mL Papain (27 mg/mL, Sigma, P3125) in L-15 medium was added to each tube. Cells were dissociated at 37°C in a heated shaker, triturating using a filtered low-binding tip every 5 min for a total of 40 min. The dissociation reaction was stopped by adding 1:1 volume of prewarmed sample buffer (1% fetal bovine serum in L-15). Cells were strained using a 40 µm nylon sterile cell strainer (Falcon, 352340) into a 50 mL tube

and transferred to a 5 mL FACS tube (Falcon, 352235). Cells were stained with fluorescent-conjugated anti-CD34-APC (Invitrogen 50-0341-80) and anti-CD31-PE (Invitrogen 12-4321-80) at 1:200 dilution. DAPI (1 mg/mL) was added (1:1000) immediately prior to FACS using the BD FACSAria sorters into 1.5 mL low-binding tubes with 100 µL of full alpha MEM culture media.

## Telocyte-macrophage co-culture

Murine bone marrow-derived macrophages were generated from bone marrow cells harvested from six wild-type animals and cultured in full alpha MEM media containing 50 ng/mL recombinant mouse macrophage colony-stimulating factor (M-CSF) protein (Biotechne-416-ML-050/CF). The cells were cultured and expanded for 3 days prior co-culture with telocytes. FAC-sorted telocytes were seeded into a 48-well plate with full alpha MEM media. After 24 hr, in the control group, macrophages with 50 ng/mL M-CSF and 1 µg/mL LPS diluted in alpha MEM were added to the telocytes. In the experimental group, macrophages with 50 ng/mL M-CSF, 1 µg/mL LPS, and 100 nM tivantinib were added to the telocytes. Cells were cultured for a further 6 days in vitro.

## Quantitative (q) RT-PCR

cDNA from RNA extracted from telocyte-macrophage co-culture were subjected to qPCR analysis with the AriaMx Real-Time PCR System (Agilent Technologies) using SYBR green and gene-specific primers. Reactions were repeated in triplicates. Relative expression levels were calculated using $2^{-\Delta\Delta CT}$ method using *Rsp19* as an endogenous housekeeping gene. Differences between experimental groups were compared using an unpaired two-tailed Student's *t*-test, and p-value ≤0.05 was considered statistically significant.

## Microcomputed tomography

Maxilla samples were fixed in 4% PFA overnight followed by three washes in PBS. Samples were scanned on a SCANCO µCT50 scanner with 70 kVp voltage and a tube current of 114 µA at 6 µm isotropic voxel size. Scans were analysed by MicroView software.

## Statistical analysis

Statistical analysis was performed using an unpaired Student's *t*-test using GraphPad Prism software. p<0.05 was considered statistically significant.

## Acknowledgements

We thank Prof. Qingbo Xu (King's College London) for kindly providing the *Cd34*<sup>CreERT2/+</sup>;*Rosa26*<sup>td-Tomato/+</sup> mice. We thank Dhivya Chandrasekaran, Fernanda Suzano, and Christopher Healy for technical assistance. We sincerely appreciate Dr Cynthia Andoniadou for her valuable comments and suggestions, which helped us to improve the quality of the manuscript. The research described was supported by the National Institute for Health Research's Biomedical Research Centre based at Guy's and St Thomas' NHS Foundation Trust and King's College London. The views expressed are those of the authors and not necessarily those of the NHS, the National Institute for Health Research, or the Department of Health. JZ was supported by the China Scholarship Council.

## Additional information

### Funding

| Funder | Grant reference number | Author |
|---|---|---|
| Chinese Academy of Agricultural Sciences | | Paul Sharpe |
| NIHR BioResoure | | Paul Sharpe |
| China Scholarship Council | | Jing Zhao |

| Funder | Grant reference number | Author |
|--------|------------------------|--------|

The funders had no role in study design, data collection and interpretation, or the decision to submit the work for publication.

## Author contributions

Jing Zhao, Data curation, Software, Formal analysis, Validation, Methodology, Writing – original draft, Writing – review and editing, Conceptualization; Anahid A Birjandi, Methodology; Mohi Ahmed, Formal analysis, Methodology, Writing – review and editing; Yushi Redhead, Data curation, Methodology; Jose Villagomez Olea, Conceptualization, Methodology; Paul Sharpe, Conceptualization, Supervision, Validation, Investigation, Writing – original draft, Project administration, Writing – review and editing

## Author ORCIDs

Jing Zhao ⓘ http://orcid.org/0000-0003-2906-0616
Paul Sharpe ⓘ http://orcid.org/0000-0003-2116-9561

## Ethics

All mouse work was approved by UK Home Office under the project license 70/7866 and P5F0A1579, approved by the KCL animal ethics committee.

## Decision letter and Author response

Decision letter https://doi.org/10.7554/eLife.72128.sa1
Author response https://doi.org/10.7554/eLife.72128.sa2

# Additional files

## Supplementary files

• Transparent reporting form

## Data availability

Sequencing data have been deposited in GEO under accession codes GSE167917 and GSE160358. Scripts used to perform analysis are available on GitHub: https://github.com/JingZhaoK/telocytes.git (copy archived at swh:1:rev:374a87d441dad264bc97e13051ce3b3de3d227f4).

The following datasets were generated:

| Author(s) | Year | Dataset title | Dataset URL | Database and Identifier |
|-----------|------|---------------|-------------|-------------------------|
| Zhao J, Sharpe P | 2022 | Telocytes regulate tissue resident macrophages in periodontal disease | https://www.ncbi.nlm.nih.gov/geo/query/acc.cgi?acc=GSE167917 | NCBI Gene Expression Omnibus, GSE167917 |
| Sharpe P | 2022 | Periodontal ligament tissue periodontitis | https://www.ncbi.nlm.nih.gov/geo/query/acc.cgi?acc=GSM5115470 | NCBI Gene Expression Omnibus, GSM5115470 |

The following previously published dataset was used:

| Author(s) | Year | Dataset title | Dataset URL | Database and Identifier |
|-----------|------|---------------|-------------|-------------------------|
| Zhao J, FAure L, Adameyko I, Sharpe PT | 2021 | Cells in healthy periodontal ligament | https://www.ncbi.nlm.nih.gov/geo/query/acc.cgi?acc=GSE160358 | NCBI Gene Expression Omnibus, GSE160358 |

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
