## [Editor Report]

This article presents valuable findings on the role of a relatively understudied cell type, telocytes, in a mouse model of periodontitis. Using single-cell RNA-seq and cellular assays, the authors present convincing evidence that telocytes signal to macrophages using HGF to shift their polarisation state from inflammatory (M1) to a more ‘tissue-remodelling’ state (M1/M2). Since periodontitis is linked to many other illnesses (e.g. rheumatoid arthritis, cardiac disease, Alzheimer's disease), new insights into the cell types that play a role in the progression of the disease are important to the field of inflammatory and chronic diseases. Future studies will need to elucidate whether telocytes play similar roles in their other niches.

---

## [Decision Letter]

**Decision letter after peer review:**

Thank you for submitting your article "Telocytes regulate macrophages in periodontal disease" for consideration by *eLife*. Your article has been reviewed by 3 peer reviewers, and the evaluation has been overseen by a Reviewing Editor and Carlos Isales as the Senior Editor. The following individuals involved in review of your submission have agreed to reveal their identity: Rio Sugimura (Reviewer #3).

Essential revisions:

After consultation with the reviewers, the consensus was that although the paper has a lot of potential, some key revisions are needed for full rigor and reproducibility. Specifically:

1. The reviewers expressed concerns about lack of a sufficient support for direct mechanistic link between HGF production and telocytes, derived solely from the cellchat analysis. Although the use of mouse genetics would be the most direct and clean way to assess this relationship, the reviewers agreed that this would be beyond the scope of a revision. However, the use of a co–culture model between the macrophages and telocytes from the periodontitis model could directly help assess (i) production of cytokines by telocytes, and (ii) direct impact on macrophages phenotypes.

2. The reviewers expressed that a better justification for the choice of flow cytometry markers would be required in a revised manuscript.

3. A better characterization of the macrophage polarization markers, by qPCR or immunostainings, would enhance the conclusions on the effects of telocyte derived HGF on macrophages.

4. For general reproducibility, the custom scripts used to perform the analysis need to be made available for review, either on a GitHub repository or as a supplementary archive (based on *eLife* guidelines). In addition, since software and database versions can impact results substantially in computational analyses, the authors need to provide version numbers for all used software and packages (e.g. R, Seurat, etc.).

5. Edits are needed for broader readership understanding and clarity (see specific reviewer comments).

*Reviewer #1 (Recommendations for the authors):*

My comments are collected whilst reading from top to bottom of the manuscript. I refer to the lines where alterations are mandatory or suggested. Before doing so: in a world that is full of articles with an ever–growing authors list, where one should openly question the real contributions of some of the authors, it is absolutely amazing that all this work has been performed by only two persons, dr. Zhao, supervised by prof. Sharpe.

– Line 111: provide details on freezing of the specimen, precautions to prevent the formation of water crystals in the frozen sections.

– Line 144: reader needs a bit more information on Cell Chat.

– Line 514, concerning the figure: proved larger font size in b for cell types. I had to blow up the figure to 200% to enable reading the cells.

– Lines 517 etc. Although specialist terms podom, podomes, piriform and moniliform were explained between lines 182–186, these terms could be reintroduced here as well, in order to facilitate the reading of the legend.

– At this stage or reading: I miss a bit the quantitative aspects. Cells shifts in percentages between perio and non–perio for all cell types, including telocytes.

– Line 197: Avoid "we asked": cells never answer, in writing nor orally. Rather: "we addressed whether…".

– Line 527: for me as an expert it is clear that the black area is tooth and partially alveolar bone. Please add white lettering T and AB for the not so experienced reader. Or even C and R for crown and root. This will provide some orientation.

– In the text, describe in more detail where the telocytes accumulated in the periodontal ligament. Is it more towards the root surface, or in the middle, or dispersed or more towards alveolar bone? In other words, not only the association with blood vessels but also anatomically.

– Based on the figure, I am not convinced that telocytes do not increase over time. 1d and 1m <2m 1y in the figure. Some counts are needed to convince.

– Line 186: To determine …. derived neural crest: One extra sentence is needed to introduce why this model is suitable for this.

– If telocytes are in proximity of blood vessels, could it be that they are mistaken for pericytes? This could be added to the discussion.

– Figure 3d: indicate control and perio in micrographs an/or in the legend. The cell morphology seems strikingly different (fibroblast shape vs. roundish) between the two micrographs. Please describe this and probably explain if possible. One could consider the proliferative nature of the telocytes in periodontitis group, which probably automatically makes them more round.

– Some quantification of the percentage and number of Ki67 positive cells is needed both for control and perio.

– Line 242 "The recipient cells are macrophages that express…" What experimental data is provided in this study? If so, refer to a figure, if not, add reference or transfer this to the discussion.

– Line 266: "These data… M1 to M2" needs absolutely more explanation. Alternatively, transfer this sentence to a bit lower, underneath the bit on c–Met.

– Line 590: Wonderful figure. I suggest to use this figure, but also to enlarge the bright yellow square of telocytes, just above the highlighted fibroblasts, including readability of the genes. Make an extra sub–figure of this part.

*Reviewer #2 (Recommendations for the authors):*

In this study Zhao et al., reveal Telocytes as the source signal for HGF which regulate macrophages behavior during periodontitis from M1(pro–inflammatory) to M2(anti–inflammatory) state. In this study Telocytes are identified as CD34+CD31– cells. The Wnt1Cre and Cd34 CreERT2 reporter mice were used to trace Telocytes. This study suggest that in homeostasis Telocytes are quiescent and are activated and proliferate following periodontitis. Telocytes are a source of HGF which in terms regulate macrophages that express the HGF receptor Met, from M1(pro–inflammatory) to M2(anti–inflammatory) state. This discovery is important and may suggest new approaches to treat periodontitis if was strongly supported by the data. In order to consider this study for publication main concerns are need to be addressed:

1. It is not clear what is the physiological significance of HGF signaling from Telocytes to periodontitis. Inhibition of HGF signaling by Tivantinib is systemic and not Telocyte specific and that result in a moderate reduction in a M2 macrophages marker Arg1. The authors should manipulate HGF signaling in Telocytes and detailed analysis of the macrophages state following inhibition is needed.

2. The authors claim that in periodontitis Telocytes proliferate while quiescent in homeostasis. This conclusion is based on CD34 and Ki67immunostaining, which is not convincing. Ki67 is a nuclear whereas CD34 a surface marker. Due to the structure of Telocytes, correlating nuclear with surface markers is challenging. A cleared whole mount staining might help, otherwise the authors might use a nuclear marker for Telocytes based on their RNAseq data and co–stain for Ki67.

3. The authors conclude that Telocytes are a neural crest derivative based on staining for CD34 ex–vivo culture cells isolated from Wnt1driven Cre reporter mouse line. In order to trace the developmental origin of Telocytes a detailed in–vivo developmental analysis is needed. This conclusion is not supported by the data and to my opinion doesn't add much insights to the study.

*Reviewer #3 (Recommendations for the authors):*

The identification of telocytes in periodontium is intriguing. However, the analysis has been rather descriptive and superficial at this stage. I encourage the authors to provide the functional significance of telocytes using mouse genetics. Then I would reconsider this manuscript. I appreciate the authors' willingness to let reviewer(s) reanalyze and validate their informatics analysis.

---

## [Author Response]

Essential revisions:After consultation with the reviewers, the consensus was that although the paper has a lot of potential, some key revisions are needed for full rigor and reproducibility. Specifically:1. The reviewers expressed concerns about lack of a sufficient support for direct mechanistic link between HGF production and telocytes, derived solely from the cellchat analysis. Although the use of mouse genetics would be the most direct and clean way to assess this relationship, the reviewers agreed that this would be beyond the scope of a revision. However, the use of a co–culture model between the macrophages and telocytes from the periodontitis model could directly help assess (i) production of cytokines by telocytes, and (ii) direct impact on macrophages phenotypes.

We thank the reviewers and editor for this suggestion which is something that will considerably strengthen the manuscript and its conclusions. We thus chose to put most of our efforts into developing methods to address this. We co-cultured FACs-sorted telocytes together with human macrophages in the presence or absence of the c-MET inhibitor tivantinib. Through qPCR analyses of marker gene expression, we show that telocytes are indeed the source of HGF that drives M1 macrophages to adopt an M1/M2 phenotype (page 17, lines 352-370). The data is presented as a new figure, Figure 6. The original Figure 5 was modified, and part of the data now appears in the new Figure 6.

2. The reviewers expressed that a better justification for the choice of flow cytometry markers would be required in a revised manuscript.

The best markers currently described in the literature to identify telocytes are by dual immunolabelling with CD34 and either CD31 (telocytes express CD34 but not CD31), c-Kit, Vim or PDGFRα (see refs 10, 12-15). We used CD34+CD31- immunolabelling in our study for FAC-sorting as telocytes, and the morphology of these cells with telocyte characteristics were confirmed in culture prior to downstream experiments. The CD34+CD31- cells were a heterogenous population as not all adopted a telocyte morphology following 24 hr in culture. Nonetheless, we obtained sufficient cells to make a significant difference in downstream qPCR experiments. We provided this explanation in page 17, lines 354-359.

3. A better characterization of the macrophage polarization markers, by qPCR or immunostainings, would enhance the conclusions on the effects of telocyte derived HGF on macrophages.

Macrophage polarisation markers described in the literature include iNOS, Arg1, CD80 and CD163, with iNOS and CD80 labelling M1 while Arg1 and CD163 labelling M2 macrophages (Jablonski et al., 2015). We included a supplementary figure (Supp.Figure 5) showing analyses of key macrophage polarisation markers by qPCR in cultured M1 and FAC-sorted telocytes. We provided this explanation in page 17, lines 359-361.

4. For general reproducibility, the custom scripts used to perform the analysis need to be made available for review, either on a GitHub repository or as a supplementary archive (based on eLife guidelines). In addition, since software and database versions can impact results substantially in computational analyses, the authors need to provide version numbers for all used software and packages (e.g. R, Seurat, etc.).

Link to GitHub repository is provided https://github.com/JingZhaoK/telocytes.git (page 22, line 436)

5. Edits are needed for broader readership understanding and clarity (see specific reviewer comments).

See specific comments for response

Reviewer #1 (Recommendations for the authors):My comments are collected whilst reading from top to bottom of the manuscript. I refer to the lines where alterations are mandatory or suggested. Before doing so: in a world that is full of articles with an ever–growing authors list, where one should openly question the real contributions of some of the authors, it is absolutely amazing that all this work has been performed by only two persons, dr. Zhao, supervised by prof. Sharpe.– Line 111: provide details on freezing of the specimen, precautions to prevent the formation of water crystals in the frozen sections.

The sucrose processing and OCT embedding is specifically designed and an established protocol for freezing, with the addition of EtOH during the embedding step to prevent bubbles forming. A brief modification is made in the methods section (page 7, line 120).

– Line 144: reader needs a bit more information on Cell Chat.

This is now on page 8, line 155. CellChat version was added and scripts provided.

– Line 514, concerning the figure: proved larger font size in b for cell types. I had to blow up the figure to 200% to enable reading the cells.

Figure 1 has now been modified to show cluster names, and f added as an additional panel.

– Lines 517 etc. Although specialist terms podom, podomes, piriform and moniliform were explained between lines 182–186, these terms could be reintroduced here as well, in order to facilitate the reading of the legend.

These terms are now included in the legend.

– At this stage or reading: I miss a bit the quantitative aspects. Cells shifts in percentages between perio and non–perio for all cell types, including telocytes.

Cell percentage show a decrease of TCs, however this can be caused by the dramatic increase in the portion of immune cells in periodontitis sample.

**Author response image 1. sa2fig1:** 

– Line 197: Avoid "we asked": cells never answer, in writing nor orally. Rather: "we addressed whether…".

Changed as suggested.

– Line 527: for me as an expert it is clear that the black area is tooth and partially alveolar bone. Please add white lettering T and AB for the not so experienced reader. Or even C and R for crown and root. This will provide some orientation.

Figures modified as suggested.

– In the text, describe in more detail where the telocytes accumulated in the periodontal ligament. Is it more towards the root surface, or in the middle, or dispersed or more towards alveolar bone? In other words, not only the association with blood vessels but also anatomically.

Telocytes are randomly dispersed during homeostasis but in periodontitis, they tend to accumulate towards the crown side of the periodontal tissue. The text has been added in Figure 2 legend.

– Based on the figure, I am not convinced that telocytes do not increase over time. 1d and 1m <2m 1y in the figure. Some counts are needed to convince.

See comments related to Fig3d below.

– Line 186: To determine …. derived neural crest: One extra sentence is needed to introduce why this model is suitable for this.

Added as suggested (page 13, line 244).

– If telocytes are in proximity of blood vessels, could it be that they are mistaken for pericytes? This could be added to the discussion.

Pericyte marker expression profile – Cd146 (MCAM), NG2, PDGFRb – is different to telocytes therefore we do not think these cells have been mistaken for each other. We added some text in page 13, lines 249-252 and Figure 1 legend.

– Figure 3d: indicate control and perio in micrographs an/or in the legend. The cell morphology seems strikingly different (fibroblast shape vs. roundish) between the two micrographs. Please describe this and probably explain if possible. One could consider the proliferative nature of the telocytes in periodontitis group, which probably automatically makes them more round.

Telocyte in the left figure is indeed also round (as indicated by an arrow). We noticed that when in homeostasis, CD34cre-labelled RFP cells are mostly expressed by endothelial cells (which express CD31 as well) with very strong RFP signal, whereas telocytes express very low levels of RFP. In this case, the RFP signal is probably too strong to see the CD31 signal in the same panel, but such bright signals are more likely to be endothelial cells rather than telocytes. Figure 2 shows the same – majority of the cells are endothelial cells with strong RFP expression unlike telocytes. In disease model, CD34cre-labelled RFP signal is lower in endothelial cells but higher in telocytes. Therefore, in the left panel, the telocyte is the dim cell indicated by the arrow, which does not express CD31.

– Some quantification of the percentage and number of Ki67 positive cells is needed both for control and perio.

Reference to Figure 4-c added, page 15, line 309.

– Line 242 "The recipient cells are macrophages that express…" What experimental data is provided in this study? If so, refer to a figure, if not, add reference or transfer this to the discussion.

Reference to Figure 4-c added, page 15, line 309.

– Line 266: "These data… M1 to M2" needs absolutely more explanation. Alternatively, transfer this sentence to a bit lower, underneath the bit on c–Met.

This sentence has been removed.

– Line 590: Wonderful figure. I suggest to use this figure, but also to enlarge the bright yellow square of telocytes, just above the highlighted fibroblasts, including readability of the genes. Make an extra sub–figure of this part.

Supp Figure 1 has been modified as suggested.

Reviewer #2 (Recommendations for the authors):In this study Zhao et al., reveal Telocytes as the source signal for HGF which regulate macrophages behavior during periodontitis from M1(pro–inflammatory) to M2(anti–inflammatory) state. In this study Telocytes are identified as CD34+CD31– cells. The Wnt1Cre and Cd34 CreERT2 reporter mice were used to trace Telocytes. This study suggest that in homeostasis Telocytes are quiescent and are activated and proliferate following periodontitis. Telocytes are a source of HGF which in terms regulate macrophages that express the HGF receptor Met, from M1(pro–inflammatory) to M2(anti–inflammatory) state. This discovery is important and may suggest new approaches to treat periodontitis if was strongly supported by the data. In order to consider this study for publication main concerns are need to be addressed:1. It is not clear what is the physiological significance of HGF signaling from Telocytes to periodontitis. Inhibition of HGF signaling by Tivantinib is systemic and not Telocyte specific and that result in a moderate reduction in a M2 macrophages marker Arg1. The authors should manipulate HGF signaling in Telocytes and detailed analysis of the macrophages state following inhibition is needed.

This experiment is not possible since it requires the cleaved form of HGF.

2. The authors claim that in periodontitis Telocytes proliferate while quiescent in homeostasis. This conclusion is based on CD34 and Ki67immunostaining, which is not convincing. Ki67 is a nuclear whereas CD34 a surface marker. Due to the structure of Telocytes, correlating nuclear with surface markers is challenging. A cleared whole mount staining might help, otherwise the authors might use a nuclear marker for Telocytes based on their RNAseq data and co–stain for Ki67.

C-kit was used in some of the TC papers, which is expressed in the nucleus, but in our seq data, it is not strongly expressed nor expressed exclusively in CD34+ TCs:

3. The authors conclude that Telocytes are a neural crest derivative based on staining for CD34 ex–vivo culture cells isolated from Wnt1driven Cre reporter mouse line. In order to trace the developmental origin of Telocytes a detailed in–vivo developmental analysis is needed. This conclusion is not supported by the data and to my opinion doesn't add much insights to the study.

We use Wnt1cre mouse to determine the origin of telocytes, reference paper of using Wnt1cre to label neural crest derived cells is added (page 13, line 244-245).